# Strategyproof Learning with Advice

**Eric Balkanski**                                    EB3224@COLUMBIA.EDU
*Columbia University*

**Cherlin Zhu**                                       CZ2740@COLUMBIA.EDU
*Columbia University*

**Editors:** Gautam Kamath and Po-Ling Loh

## Abstract

An important challenge in robust machine learning is when training data is provided by strategic sources who may intentionally report erroneous data for their own benefit (Caro and Gallien, 2010; Caro et al., 2010). A line of work at the intersection of machine learning and mechanism design aims to deter strategic agents from reporting erroneous training data by designing learning algorithms that are strategyproof (Dekel et al., 2010; Perote and Perote-Pena, 2004; Chen et al., 2018; Cummings et al., 2015; Meir and Rosenschein, 2011; Meir et al., 2008, 2010, 2012; Hardt et al., 2016; Ghalme et al., 2021; Dong et al., 2018; Ahmadi et al., 2021). Strategyproofness is a strong and desirable property, but it comes at a cost in the approximation ratio of even simple risk minimization problems.

A recent line of work on mechanism design with advice has shown that side information can be leveraged to overcome worst-case bounds in mechanism design. Strategyproof mechanisms that achieve an improved approximation ratio when the advice is accurate (consistency) and an acceptable approximation ratio when the advice is inaccurate (robustness) have been designed for problems such as strategic facility location (Agrawal et al., 2022; Xu and Lu, 2022; Balkanski et al., 2024a), auction design (Lu et al., 2024; Caragiannis and Kalantzis, 2024; Balkanski et al., 2024b), strategic scheduling (Balkanski et al., 2023), strategic assignment (Colini-Baldeschi et al., 2024), and metric distortion (Berger et al., 2024). In this paper, we study strategyproof learning in two settings: regression and classification; we provide the first non-trivial consistency-robustness tradeoffs for both.

In strategic learning, each agent $i \in \{1, \ldots, n\}$ reports a set $S_i = \{(x_{i,j}, y_{i,j})\}_j$ of labeled data points to the learner. The points $x_{i,j}$ are public information, but the labels $y_{i,j}$ are private information that agent $i$ can potentially misreport. The goal of the mechanism is to learn a function $f$ from a function class $\mathcal{F}$ that minimizes the global risk $R(f, S) = \frac{1}{|S|} \sum_{i=1}^{n} \sum_{j=1}^{|S_i|} \ell(f(x_{i,j}), y_{i,j})$ for some loss function $\ell$. The agents are strategic and aim to minimize their personal risk $R_i(f, S) = \frac{1}{|S_i|} \sum_{j=1}^{|S_i|} \ell(f(x_{i,j}), y_{i,j})$. We augment the problem of strategic learning with a potentially erroneous advice $\tilde{f} \in \mathcal{F}$ about the global risk minimizer that is given as input to the mechanism, in addition to the labeled data reported by the agents.

We first consider regression problems with the absolute loss function $\ell(x, y) = |x - y|$. For the classes of constant and homogeneous linear functions, Dekel et al. (2010) gave deterministic and group-strategyproof mechanisms that achieved 3 approximation to the minimum global risk and showed that these guarantees were tight. We provide the following results for regression over constant functions:

- Introduce a deterministic and strategyproof mechanism that is, for any $\gamma \in (0, 2]$, $1+\gamma$ consistent and $1 + 4/\gamma$ robust,

- Show that this mechanism is group strategyproof when agents have unique personal risk minimizers, and

- Prove that the consistency-robustness tradeoff is tight under mild assumptions.

This mechanism and its guarantees also extend to homogeneous linear functions. We then consider binary classification problems over the 0-1 loss function in the shared input setting, where agents have identical points but may disagree on their labels, and the function class is a set of specific labelings for the points. Meir et al. (2012) give a deterministic mechanism that achieves a $2n - 1$ approximation and show that no deterministic mechanism can achieve a sublinear approximation. They also give a 3 approximate randomized mechanism, which they also show to be tight. We provide the following results:

- Any deterministic and $o(n)$ consistent mechanism has unbounded robustness, and

- Any random mechanism with consistency better than 3 is $\Omega(n)$ robust.

For the special case of function classes with only two labelings, we extend the deterministic mechanism for regression and its guarantees, as well as provide a randomized mechanism parametrized by $\gamma \in (0, 1]$ that is $1 + \gamma$ consistent and $1 + 1/\gamma$ robust.[1]

**Keywords:** Mechanism design, algorithms with predictions, strategyproof learning

## Acknowledgments

We thank the anonymous reviewers for their constructive comments. The work of E. Balkanski and C. Zhu was supported in part by the National Science Foundation [Grants CCF-2210501 and IIS-2147361].

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
