# OpenReview forum: "Strategyproof Learning with Advice"
_algorithmiclearningtheory.org/ALT/2025/Conference — ALT 2025_

### Official Review · Reviewer_QqoV · 2024-11-09
**Initial review**

**Rating:** 8
**Confidence:** 3

**Review:**

The paper studies the following strategic learning problem. There are n agents, each agent $i$ holds a set of labeled data $\{(x_{ij},y_{ij})\}$. The agents report a possibly manipulated dataset (with labels being misreported) to the learner, and the learner learns a function $f$ in a given class that minimizes the overall loss $\sum \ell(f(x_{ij}),y_{ij})$. Each agent is strategic, and wants to minimize its own total loss.

The paper studies the strategic learning problem with advice. In addition to the labeled dataset, the learner also receives an advice function $\tilde{f}$. The paper shows that when the function class is constant functions or linear functions, there is a deterministic strategy-proof mechanism that is $(1+\gamma)$-consistent (which corresponds to the competitive ratio when the advice is a true minimizer) and $(1+4/\gamma)$-robust (which corresponds to the worst-case competitive ratio for arbitrary advice) for any $\gamma\leq 2$, and the parameters are tight. For general classification problems, when there are 2 labels, the previous constant function result generalizes. When there are multiple labels, large robustness parameter is needed for both deterministic and randomized mechanisms.

Evaluation: The paper generalizes the learning with advice type problems to the strategic learning setting, and provides tight consistency/robustness characterization of strategy-proof mechanisms with advice for constant functions and linear functions. For classification problems, the paper provides strong hardness results on robustness. Thus, the paper provides a clear and complete solution to a natural problem and is well above the bar of the conference.


Minor typo:
Page 4. “Ouput” -> “Output”

**Paper Award:**

No

---

### Official Review · Reviewer_mHbv · 2024-11-13
**Very interesting problem, strong results**

**Rating:** 7
**Confidence:** 3

**Review:**

**Summary**

The paper studies strategyproof learning problems in the setting of algorithms with predictions.  In strategy-proof learning, agents might manipulate some data labels for their own benefit, and the objective of the algorithm is to be robust in the face of these manipulations. The authors assume that the algorithm is given an advice on the optimal mapping $f$, belonging to a class $\mathcal{F}$, between the data space and the labels space, that minimizes the global risk. The objective then is to use this advice to have a consistent and robust algorithm for learning $f$.
The authors first focus on regression, with $\mathcal{F}$ restricted to the set of constant functions, and they design a deterministic algorithm with an optimal consistency-robustness tradeoff.
Then, for classification, they prove impossibility results in the case of three or more labels, then they design deterministic and randomized learning-augmented algorithms for the case binary classification.


**Strengths**

- The studied problem is very interesting. Many decision-making problems have been studied in the learning-augmented setting, but there haven't been much work considering learning problems in this framework, and none (to my knowledge) has studied strategyproof learning.
- The paper is well-written, and the authors give intuitions on their algorithms and proofs.
- The paper explores multiple learning problems (regression, binary classification, classification with more than two labels), and provides strong results.
- The consistency-robustness tradeoff for regression with constant functions is optimal, and setting $\gamma = 2$ gives the optimal approximation ratio without advice.
- The authors also give smoothness guarantees, showing how the consistency and robustness bounds are interpolated as the advice error becomes more important.

**Weaknesses**

- The authors do not provide any experimental results supporting the practicality of their algorithms.
- For regression, the authors only consider constant functions, which is very limited.
- Since the paper focuses on "learning with advice," maybe the authors should cite papers studying similar problems, such as "Online Classification with Predictions" (Raman, Tewari) or "Learning-Augmented k-means Clustering" (Ergun, Feng, Silwal, Woodruff, Zhou).
- I am unsure how the advice on the optimal function $f$ can be obtained in practice. The authors mention that it is weaker than having advice on the true labels, but I don't see how this can be obtained either.

**Paper Award:**

No

---

### Official Review · Reviewer_VQ4U · 2024-11-16
**Review on strategyproof learning with advice**

**Rating:** 7
**Confidence:** 3

**Review:**

This paper deals with the question of designing strategyproof learning algorithms in a regression or classification setting where an additional prediction on the optimal solution can be leveraged to improve the mechanism performance. The learning mechanism is required to be strategyproof, in that no agents can benefit from misreporting their data. Building on previously used strategyproof mechanisms, new mechanisms which incorporate the advice are proposed.  In line with the literature on algorithms with predictions, worst-case trade-offs between robustness and consistency are derived, and this trade-off is shown to be tight in both the regression and two-labellings classification setting.

Strengths:
- The results of the paper are complete, tight, and cover multiple settings previously studied in strategy proof learning
- The results and setting are novel
- The Lower-bound to show tightness in the median estimation setting is non-trivial

Weaknesses
- The writing in some parts could be improved, see comments
- I feel that the originality/contribution of those new results are limited, in that it is a standard mechanism + predictions problem

I have quickly read through most of the proofs for the median estimation setting, and they seem correct, besides some small omissions.

Some comments on the writing:
- The introduction and related work section should be more developed and better highlight why one should care about strategy-proof learning with predictions in particular. Right now it feels like a juxtaposition of a list of papers from strategy-proof learning and then papers on Mechanisms with advice. The question of combining the two is indeed natural from a theoretical standpoint, but it should still be better motivated; as an example the real-world application mentioned at the beginning is the same as the one in Dekel et al (2010) dealing with strategyproof regression.
- In terms of related work, I think it should be better explained what were the methods in previous works, to compare and see what techniques were used in this paper. For instance, I don’t think that it was explicitly mentioned that the Project-and-fit algorithm is taken from Dekel et al (2010), although the results are indeed cited. Similarly a common technique in algorithms with predictions is to take the convex combination of the prediction and of the ‘standard’ algorithm and then analyze the performance. Here a similar method is used in that somewhat a convex combination of the true data-set and the predicted data set is used.
- Some parts of the proofs or arguments are confusing and should be better explained. For instance in the first paragraph of the proof of Lemma 5, it is not clear to me what is the relationship between Observation 2 and the statement right after it. After reading the original strategyproof argument in Dekel et al I understand why, but on a first read it was quite unclear. Similarly in the paragraph below Theorem 16 the role of the $g_i$ was unclear. Is it correct that the $g_i$ are basically a way to get instances of agents data such that the order induced by the risk is consistent with a specific preference profile over labellings?
- There are Claims, Observations, Lemma, but I don’t really understand the difference between them honestly. Why is Lemma 9 not a claim or an observation for instance?

Questions:
- Can we say anything about the smoothness of the mechanism with respect to the predictions? That is to say, what is the best value of $\beta$ such that $R(M(S,\tilde{a}),S)< \beta R(\tilde{a},S)$?
- In Dekel et al group strategy-proofness was proven for any size of the data-sets. Here $\vert S_i \vert$ is required to be odd for all $i$. Is it a limitation of the proof, the mechanism, or the setting with advice? More precisely, are there examples which show that if $S_i$ is even then it cannot be group-strategy proof for any mechanisms? If not, can either the proof or the mechanism be fixed?
- Do the results hold if $\mathbb{Y} \subset \mathbb{R}^2$?
- Do any lower bound results similar to Theorem 7 hold if we consider randomized mechanisms for the median estimation problem?
- Proof of Lemma 10: I don’t understand why the application of Corollary 25 immediately leads to the last inequality on top of page 21 to be greater than $1+\gamma$.
- Why can we assume in the proof of Claim 12 that $D’>b$?

Small comments/typos:
- Consistency and robustness are mentioned in Theorem page 2, but are not defined at this point. Maybe a brief explanation in words should be added.
- Page 2: “of the advice error $\eta$.” -> “of the advice error.”
- Page 3: “for the special case of classes” -> “for the special case of function classes”
- Why not scale gamma in the regression setting so that it is always in $[0,1]$ to be consistent with the classification setting?
- In table 1, the $O(n)$ should be replaced by $\Omega(n)$, same in the Theorem 17. In table 1 and Theorem 15, the robustness is not unbounded: It is lower bounded by $\Omega(n)$.
- Lemma 2: Missing space before parenthesis
- Theorem 6: missing link to the proof in the appendix
- Last inequality of the proof of Lemma $2$ is only valid because $\vert y_{i,j} <a \vert \geq \vert y_{i,j} \geq a \vert$ as $a^*$ is the median and $a>a^*$.
- Proof of Lemma 4 end of first paragraph: missing something like “And because R(a,S) is decreasing over $[b,a^*]$...”
- Claim 12 is proven before Claim 11
- Proof of Claim 11: M is used instead of D

---

I thank the authors for their reply.

I am still between a 6 and 7. It would be better to have a smoothness guarantee rather than a consistency one, which is quite weak. Nonetheless due to the lower bound that seems non-trivial to obtain, and the fact that the results are quite complete (both median estimation and classification) I am going to change my score to a 7, provided that the authors do make the necessary writing improvements mentionned by another reviewer and I.

**Paper Award:**

No

---

### Official Review · Reviewer_bvB7 · 2024-11-17
**Clear paper, comments on writing & definitions**

**Rating:** 6
**Confidence:** 3

**Review:**

This paper considers the setting of learning in the presence of strategic agents who are in charge of labelling datapoints, while making using of "advice" in the form of a given fixed hypothesis. The layout of interests are as follows: each agent wants the final hypothesis to produce the smallest loss on its own subpopulation, while the designer would like to minimize loss on the total population. For the task of learning constant functions against absolute loss, the work provides an algorithm that trades off consistency (following advice when it is good) and robustness (being within a certain factor of optimal); at the extreme matching Dekel et al. The algorithmic idea does not defy expectations, i.e., interpolate these considerations when selecting the final hypothesis; but since the obtained trade-off is tight the result itself is interesting. Another main result is that in the 0/1 classification setting (if agents share datapoints), advice to the learner can't be used to mediate robustness-consistency tradeoffs any better than ignoring the advice altogether.

I found the mathematical parts to be cleanly written and linearly readable. This is a positive.

One worry I have is that (only) for upper bounds given the definition of consistency, we might not be asking enough from the algorithm. For example a natural notion would be $Cost(\mathcal{A}) \leq \min\\{\alpha Cost(\widetilde{f}), \beta Cost(f^*)\\}$. Here, the consistency bounds would come into play even if the advice is imperfect, but good. The current definition on page 5 implies consistency is only up for consideration when the advice is exactly correct.

Comments on writing:
I should note that I had to read Dekel et al to understand why this particular arrangements of interest is well motivated, and the incentive-compatible ML literature has evolved quite a bit in the past 10 years to settings involving stronger conflicts of interests. In any case, I hope that the authors can include a paragraph (e.g., mirroring examples in Dekel et al) to motivate the setting. Similarly, briefly defining consistency and robustness in words before the theorem statement on page 2 would help, although this I feel less strongly about, because these definitions (almost; see above) match what an uninitiated reader (like me!) would imagine them to be.

The other bit that would help in making a stronger case for this paper is to expose tools and directions that the follow-up works could build on independently of the end results. For example, is there a "core" that although central to the lower bound construction when assumed away could result in nice algorithms?

Finally, I'm divided between assigning a 6 or a 7; so I'm open to persuasion both by the authors and by what the other reviewers have to say.

Edits: fixed typos.

---
I think post rebuttal I would like to maintain my evaluation. The authors did not present any evidence to assuage my concern that the consistency requirements imposed on the learner are very weak (only come into effect when the advice is near? perfect); I see that another reviewer shares this concern. Neither did the authors expose a reasonable way forward in the classification setting, using insights gathered from their lower bounds. So I'm still between a 6 and a 7, leaning 6.

**Paper Award:**

No

---

### Author Rebuttal · Authors · 2024-11-24

Reviewer bvB7:

We thank the reviewer for their careful and overall positive review.

Comment regarding consistency-like bounds for imperfect but good advice, such as 𝐶𝑜𝑠𝑡(𝐴)≤min{𝛼𝐶𝑜𝑠𝑡(𝑓(pred)),𝛽𝐶𝑜𝑠𝑡(𝑓(opt))}.

A: We provide approximation guarantees that degrade as a function of the advice error, which gives stronger guarantees than consistency by itself (the definition of the advice error is given right before Theorem 6). Theorems 6 and 14 give improved guarantees as a function of the advice error for when the advice is good but not exactly correct. Your example suggestion is an interesting alternative measure for the performance of a mechanism when there is some small error in the advice.

Comment regarding motivating incentive-compatible ML.

A: In the introduction, we discuss a similar retail motivating application to the one provided by Dekel et al. as our main motivating application. We will expand the discussion of this retail application to better explain why this particular arrangement of interest is well-motivated. Thank you for the suggestion.

Comment regarding defining consistency and robustness above the theorem on page 2.

A: Thank you for pointing this out, we will add the following informal statement above the theorem: We say a mechanism is alpha consistent if it achieves an alpha approximation when the prediction is equal to the optimal, i.e., risk minimizing, function, and we say a mechanism is beta robust if it achieves a beta approximation given any arbitrarily bad prediction.

Question: The other bit that would help in making a stronger case for this paper is to expose tools and directions that the follow-up works could build on independently of the end results.

A: Given a dataset of labeled points ${(x_i, y_i)}_i$, this paper focuses on strategic learning problems where the agents can misreport the labels $y_i$. A direction that follow-up works could build on independently that we find interesting and promising is to consider problems where the agents can misreport the points $x_i$ instead of the labels $y_i$. Such problems are also well-studied in the strategic learning literature and it would be interesting to study them with predictions. We will mention this direction as future work.

---

> ### Author Rebuttal · Authors · 2024-11-24
>
> Reviewer VQ4U (pt1):
>
> We thank the reviewer for their careful and overall positive review.
>
> Comment: I feel that the originality/contribution of those new results are limited, in that it is a standard mechanism + predictions problem
>
> A: We view strategic learning as an important area at the intersection of mechanism design and learning theory. Our main conceptual contribution is showing that side advice can be helpful for strategic regression but that, in general, it is not helpful for strategic classification if we wish to maintain robustness guarantees. We believe this nuanced view on the benefits of side advice provides a contribution that goes beyond a standard mechanism + prediction result.
>
> Comment regarding motivating strategyproof learning with predictions:
>
> A: Thank you for this suggestion. The goal of the real-world retail application mentioned at the beginning was exactly to highlight why one should care about strategyproof learning with advice, and the reviewer is correct that this application is the same as the one in Dekel et al. We will develop how this example application connects, and is mentioned, in the existing work on strategyproof learning and on mechanism design. We will also develop how strategyproof learning with advice is important beyond this example application.
>
> Comment: it should be better explained what were the methods in previous works [...] the Project-and-fit algorithm is taken from Dekel et al (2010) [...]
>
> A: The reviewer is correct that our mechanism is related to the one introduced by Dekel et al. In particular, when $\gamma = 2$, our mechanism ignores the predictions and corresponds exactly to their mechanism. We should have highlighted this and will make sure to do that in the next version of the paper. The main difference is how we leverage the predictions in the case $\gamma < 2$.
>
> Comment: A common technique in algorithms with predictions is to take the convex combination of the prediction and of the ‘standard’ algorithm [...]  Here a similar method is used in that somewhat a convex combination of the true data-set and the predicted data set is used.
>
> A: We agree that our mechanism combines the true dataset and the prediction, but conceptually this combination differs from the “convex combination of the prediction and of the ‘standard’ algorithm” mentioned by the reviewer. In particular, we highlight that this convex combination with constant weight on the prediction would fail and have unbounded robustness.

---

> > ### Author Rebuttal · Authors · 2024-11-24
> >
> > Reviewer VQ4U (pt2):
> >
> > Comment: not clear to me what is the relationship between Observation 2 and the statement right after it.
> >
> > A: We will provide the following intuitive explanation of what Observation 2 represents: This observation shows that, amongst the union of projected points S’ and $\lambda |S|$ copies of $\tilde{a}$, at least half of these points fall on either side of the constant returned by PFA.
> >
> > Comment regarding the role of $g_i$.
> >
> > A: That is correct, and we will add this explanation to that paragraph.
> >
> > Comment regarding the difference between Claims, Observations, Lemmas.
> >
> > A: We use observations for facts that only require a one-sentence explanation instead of a full formal proof. Claims are statements within the proofs of larger results whose proofs are deferred to the appendix. We understand that this may be confusing and we will extract claims and turn them into lemmas in future submissions. We also agree that Lemma 9 could instead be an observation. Thank you for raising these points.
> >
> > Comment: smoothness of the mechanism with respect to the predictions? [...] best value of 𝛽 such that $R(M(S, \tilde{a}), S) < \beta R(\tilde{a}, S)$?
> >
> > A: This definition of 𝛽 is a very interesting alternative metric to the prediction error we define in the paper. In fact, this is an extension that we were thinking of studying in future work. Our results do not easily extend to obtain such smoothness guarantees.
> >
> > Comment: Do the results hold if 𝑌⊂𝑅2?
> > A: To answer this question, we first need to define the loss function between two 2-D points. If $L_1$ loss is used, then our results generalize to $R^2$, and in fact to arbitrary $R^k$, because we can separately select the constant value in each dimension separately by using our mechanism for 1-D. This does not feel like a particularly useful generalization to $R^2$. If instead, for example, we use $L_2$ loss, then it is unknown what guarantees our mechanism achieves, and that may be an interesting direction to pursue.

---

> > > ### Author Rebuttal · Authors · 2024-11-24
> > >
> > > Reviewer VQ4U (pt3):
> > >
> > > Comment: In Dekel et al group strategy-proofness was proven for any size of the data-sets. Here |𝑆𝑖| is required to be odd for all 𝑖. Is it a limitation of the proof, the mechanism, or the setting with advice?
> > >
> > > A: We can show that for both our mechanism and the one presented in Dekel et al., group strategyproofness does not hold when agents do not have single-peaked preferences (see end of answer for proof). We enforce single-peakedness by using the sufficient condition of having an odd number of points for each agent, but single-peakedness itself can also be sufficient when proving group strategyproofness (see, e.g.,  the paper ``Strategyproof Linear Regression in High Dimensions” by Chen et al. that assume single-peakedness for their group-strategyproofness guarantees). We do not have a proof that shows that “if 𝑆𝑖 is even then it cannot be group-strategy proof for any mechanism”, but this would be an interesting result to try to prove.
> > >
> > > Consider the following instance of agent types which is a counterexample to project-and-fit from Dekel et al. being groupstrategyproof when agents can have single-plateau preferences: agent 1: (0, 0), agent 2: (0, 1), agent 3(1, 1). Observe that $ERM(F, (0,1)) = a \in [0,1]$, and the mechanism will output a. If a<1, then agent 2 can misreport their type as (1,1) so that the mechanism outputs 1, which strictly benefits agent 3 while not decreasing the risk of agent 2. Similarly, if a > 0, then agent 2 can misreport their type as (0,0) so that the mechanism outputs 0, which strictly benefits agent 1 without harming agent 2.
> > >
> > > Comment: Do any lower bound results similar to Theorem 7 hold if we consider randomized mechanisms for the median estimation problem?
> > >
> > > A: Theorem 7 only holds for deterministic mechanisms. An interesting direction for future work would be to study if allowing for randomization allows for better guarantees.
> > >
> > > Comment: don’t understand why the application of Corollary 25 immediately leads to the last inequality on top of page 21
> > >
> > > A: Corollary 25 states that if the mechanism returns a constant greater than $(1-\delta)z(i+1)$ given instance $S(n, k+1, t, z(i+1))$, it cannot be $1+\gamma$ consistent. It’s important to note, and this is a note that we will add, that the $\delta$ in Corollary 25 is the same $\delta$ used in Lemma 24, and therefore the proof of Lemma 10.

---

> > > > ### Author Rebuttal · Authors · 2024-11-24
> > > >
> > > > Reviewer VQ4U (pt4):
> > > >
> > > > Comment: Why can we assume in the proof of Claim 12 that 𝐷′>𝑏?
> > > >
> > > > A: In general we cannot, thank you for catching this missing case. The statement of Theorem 8 is missing a very mild assumption to handle this case, which is that the agents’ type space (i.e., their labels) and the outcome space (i.e., the constant function) are upper bounded by some arbitrarily large value T. Then we can take D’ to be equal to T, and therefore $D’ \geq b$ must hold, allowing the rest of the proof to hold.
> > > >
> > > > Comment: Consistency and robustness are mentioned in Theorem page 2, but are not defined at this point. Maybe a brief explanation in words should be added.
> > > >
> > > > A: We will add the following informal definition right before the Theorem on page 2: We say a mechanism is alpha consistent if it achieves an alpha approximation when the prediction is equal to the optimal, i.e., risk minimizing, function, and we say a mechanism is beta robust if it achieves a beta approximation given any arbitrarily bad prediction.
> > > >
> > > > Comment: Why not scale gamma in the regression setting so that it is always in [0,1] to be consistent with the classification setting?
> > > >
> > > > A: We chose the parameter gamma to be such that we achieve a $1 + \gamma$ consistency in every setting. In other words, gamma always represents the loss in consistency. This interpretation would not hold anymore if we scaled gamma as suggested.
> > > >
> > > > Comment: In table 1, the 𝑂(𝑛) should be replaced by Ω(𝑛), same in the Theorem 17. In table 1 and Theorem 15, the robustness is not unbounded: It is lower bounded by Ω(𝑛).
> > > > A: Thank you for catching this, we will switch O(n) to $\Omega(n)$, and for general classification say o(n) consistency and bounded robustness. This is because Theorem 15 states that a mechanism that achieves bounded robustness (in other words return the correct classification when every point is labeled the same by all agents) cannot achieve sublinear consistency.
> > > >
> > > > Comment: In regards to the rest of the small comments:
> > > >
> > > > A: Thank you for pointing these out, we agree with all of them and will fix them as suggested.

---

> > > > > ### Author Rebuttal · Authors · 2024-11-24
> > > > >
> > > > > Reviewer mHbv:
> > > > >
> > > > > We thank the reviewer for their careful and overall positive review.
> > > > >
> > > > > Comment: For regression, the authors only consider constant functions, which is very limited.
> > > > >
> > > > > A: This comment is not entirely correct. For regression, our results for constant functions extend to homogeneous linear functions (see Section 3.3). We will add these results to Table 1.
> > > > > For beyond constant and homogeneous linear functions, we note that the classes of functions we consider are the classes of functions for which approximation guarantees have been achieved in prior work on strategic learning (see e.g. references  [Dekel et al.,  Perote and Perote-Pena, Chen et al., Cummings et al., all five Meir et al. works, Hardt et al., Ghalme et al., Dong et al., Ahmadi et al.]). Strategic learning is a challenging problem even for these classes of functions. Studying more complex families of functions is an interesting and ambitious direction for future work where the first step is to achieve approximation guarantees without predictions.
> > > > >
> > > > > Comment: Since the paper focuses on "learning with advice," maybe the authors should cite papers studying similar problems, such as "Online Classification with Predictions" (Raman, Tewari) or "Learning-Augmented k-means Clustering" (Ergun, Feng, Silwal, Woodruff, Zhou).
> > > > >
> > > > > A: Yes, we agree with the review, thank you for the suggestion. We will incorporate these into the related works section.
> > > > >
> > > > > Comment: I am unsure how the advice on the optimal function 𝑓 can be obtained in practice. The authors mention that it is weaker than having advice on the true labels, but I don't see how this can be obtained either.
> > > > >
> > > > > A: A main potential approach to obtaining the advice would be to identify a function f that performs well on historical data from past instances, for example historical sales data in the Zara retail application scenario. Alternatively, it could also be advice provided by an expert or experienced practitioner.

---

> > > > > > ### Author Rebuttal · Authors · 2024-11-24
> > > > > >
> > > > > > Reviewer QqoV:
> > > > > >
> > > > > > Thank you for the favorable review! We will fix the typo.

---

### Meta-Review · Area_Chair_rAEk · 2024-12-05

**Recommendation:** Accept
**Confidence:** 5

**Metareview:**

This paper looks at the now quite trendy research field of learning with prediction.

All the reviewers are positive about the paper, even though the writing could be improved, especially to better reflect the actual contributions. Now that the paper is accepted, no need to oversell !

Please take into account the different reviews in your revision

**Paper Award:**

No